# High-density perimetry in the assessment of foveal avascular zone and macular structure in glaucoma

**Gustavo Coelho Caiado**⊙, **Gustavo Albrecht Samico**⊙, **Gilvan Vilarinho da Silva Filho,
Sergio Henrique Teixeira, Tiago Santos Prata, Carolina Pelegrini Barbosa Gracitelli,
Augusto ParanhosJr**⊙ *

Department of Ophthalmology and Visual Science, Glaucoma Service, Federal University of São Paulo,
São Paulo, São Paulo, Brazil

* augusto.paranhos@gmail.com

## Abstract

### Purpose

To evaluate the association between foveal avascular zone (FAZ) parameters (area, perimeter and circularity) with macular high density perimetry (Octopus Macular program), macular vessel density (mVD) and ganglion cells layer thickness (GCLT) in glaucoma patients.

### Methods

This cross-sectional study included 89 eyes from 57 glaucoma patients. All participants underwent high-density perimetry (Octopus 900 Macular and G programs). FAZ metrics and mVD were obtained via Optical Coherence Tomography Angiography (OCTA) (Triton, Topcon), and GCLT was measured across global, superior, and inferior sectors. FAZ parameters were automatically extracted using ImageJ with axial length correction. Structure–structure and structure–function associations were assessed using mixed-effects linear regression models, adjusting for inter-eye correlation, age, and GCLT when appropriate.

### Results

Mean age was $66.67 \pm 7.49$ years old. FAZ area and perimeter were significantly associated with mean defect of macular program (mMD), central mVD, and GCLT ($p < 0.05$). FAZ area remained significantly associated with GCLT even after controlling for age ($p < 0.05$). FAZ perimeter and circularity were independently associated with age. FAZ perimeter and circularity were mainly age-related, while area was linked to glaucomatous damage. FAZ parameters were more strongly related to superior than inferior GCLT. Glaucoma severity was categorized using Brusini's

**Data availability statement:** The minimal anonymized dataset underlying our findings has been deposited in Zenodo and is publicly available at: https://doi.org/10.5281/zenodo.17000133.

**Funding:** The author(s) received no specific funding for this work.

**Competing interests:** The authors have declared that no competing interests exist.

Glaucoma Staging System. Higher mMD values were observed in more advanced stages of glaucoma.

## Conclusions

FAZ morphology reflects both glaucomatous damage and age-related vascular changes. By controlling for confounding variables, this study reinforces the role of FAZ metrics as complementary biomarkers for assessing structure-function relationships in glaucoma. High-density macular perimetry further improves spatial correspondence with anatomical alterations.

## Introduction

Glaucoma is a chronic, progressive, potentially blinding, irreversible eye disease causing optic nerve rim and retinal nerve fiber layer (RNFL) loss with related visual field (VF) defects [1]. Many patients remain undiagnosed or continue to progress despite currently available diagnostic and therapeutic modalities [2]. Abnormalities in retinal microcirculation and impairment of ocular blood flow have an impact on development of glaucoma [3,4]. Previous studies found that vessel densities (VD) of the optic nerve head (ONH), peripapillary area and macula have significant associations with structural glaucomatous damage expressed in the neuroretinal rim area, RNFL thickness and ganglion cell complex (GCC) thickness measured by optical coherence tomography (OCT) [5,6].

The foveal avascular zone (FAZ) is the capillary-free area in the central macula. Various methods have been used to evaluate the retinal vasculature such as fluorescein angiography [7]. OCT angiography (OCTA) is a noninvasive imaging modality that measures blood VD at the macula by detecting red blood cell movement [8] with great precision and repeatability [9–12]. Previous authors showed that reduced VD, larger FAZ area and lower FAZ circularity in the superficial retinal capillary network are associated with glaucoma [6,13,14]. Decreased macular VD, increased FAZ perimeter and decreased FAZ circularity index were also observed in eyes with glaucoma using OCTA [5]. Previous evidence indicates that alterations in macular microcirculation are closely linked to VF impairment [15–17]. In particular, Zabel et al. [15]demonstrated the association between microvascular damage in the macular superficial vascular plexus and the reducedretinal sensitivity measured by microperimetry was stringer than the correlations observer with peripapillary RNFL thickness or conventional standard automated perimetry (SAP) parameters [15].

The aim of our study was to evaluate the association between FAZ parameters (area, perimeter and circularity) with macular high density perimetry (Octopus Macular program – Octopus 900, Haag-Streit, AG, Koeniz, Switzerland), macular vessel density (mVD) and ganglion cell layer thickness (GCLT) in glaucoma patients. Through the integration of high-density macular perimetry, axial length-corrected OCTA imaging, and multivariate analytical approaches, we found that both FAZ area and perimeter were significantly correlated with macular VF sensitivity and GCLT,

especially within the superior macular region. Furthermore, we observed that FAZ perimeter and circularity were independently associated with age, indicating that FAZ geometry may be shaped by a combination of glaucomatous injury and age-related microvascular changes. The consistent structure-structure and structure-function associations observed in this study emphasize the clinical relevance of FAZ metrics as complementary biomarkers for glaucoma assessment.

## Materials and methods

### Study population

This cross-sectional study of glaucoma subjects was approved by the Ethics Committee of Federal University of São Paulo (Ethics Committee Approval Number: 35713820.2.0000.5505). Written informed consent was obtained from all participants. The patient recruitment period started on April 27, 2021, and ended on June 23, 2023. 89 eyes from 57 patients were included. Glaucomatous subjects were recruited from glaucoma division from Ophthalmology and Visual Sciences Department at Federal University of São Paulo. Patients with glaucoma were enrolled in the study based on clinical findings consistent with glaucomatous optic neuropathy, such as vertical cup-to-disc ratio >0.6, asymmetry of cup-to-disc ratio >0.2 between eyes, and presence of localized RNFL or neuroretinal rim defects corresponding to abnormal VF in SAP in the absence of any other abnormalities that could explain the findings on fundus examination [18]. Abnormal VF was defined as follows [19]: (1) outside normal limit on glaucoma hemifield test or (2) three abnormal points with P less than 5% probability of being normal or one abnormal point with P less than 1% by pattern deviation, or (3) pattern standard deviation (PSD) of 5% if the VF was otherwise normal, confirmed by two consecutive tests. A VF measurement was considered to be reliable when false-positive results were less than 15%, false-negative results were less than 15% and fixation losses were less than 20%. There was no washout period before the exams due to the severity of the glaucomatous damage. IOP was measured while the patients were using their antiglaucoma medications and was measured on the day of the exams.

All participants underwent a comprehensive ophthalmic examination, which included assessment of visual acuity, IOP measurement by Goldmann applanation tonometry, slit lamp examination of the anterior segment, fundoscopy, gonioscopy, axial length measurement by optical biometer (IOL Master 500, Carl Zeiss Meditec, Dublin, CA) and swept-source OCT (DRI OCT Triton, Topcon Inc, Tokyo, Japan) and OCTA (DRI OCT Triton, Topcon Inc, Tokyo, Japan) examination. We used the same DRI OCT to perform OCT and OCTA scans. Patients underwent standard VF examination by the static automated white-on-white Macular and G Program (Octopus 900, Haag-Streit, AG, Koeniz, Switzerland). Glaucoma severity was categorized using Brusini's Glaucoma Staging System, which classifies damage based on mean defect (MD) and PSD obtained from perimetric testing.

Participants were excluded if they were younger than 18 or older than 80 years, had a spherical equivalent outside range of + 3 to − 6 diopters, or presented with a best-corrected visual acuity worse than 1.0 logMAR. Eyes with non-glautomatous optic neuropathies, significant retinal disease (such as diabetic or hypertensive retinopathy, uveitis, age-related macular degeneration, pathological myopia with secondary macular changes, retinal detachment, or ocular trauma) were not included. Subjects with relevant neurological disorders (e.g., Parkinson's disease, Alzheimer's disease, or history of stroke) were also excluded. Additional exclusion criteria were unreliable VF examinations and OCTA scans of insufficcient quality, as defined in the image processing section.

### OCT and OCTA imaging acquisition

All subjects were examined using the macular 4.5x4.5mm scanning protocol (DRI OCT Triton, Topcon Inc, Tokyo, Japan). Topcon OCTA instrument uses a wavelength of 1050 nm with A-scan rate of 100,000 scans per second. The instrument produces maps using OCTA ratio analyses (OCTARA), which is an amplitude-decorrelation ratio-based algorithm [20]. The system automatically divided the macula into four layers, and the selected layer was superficial retinal capillary plexus (SCP). The SCP is defined as from the inner border of the RNFL to 15.6 μm from the boundary between the inner

plexiform layer and the inner nuclear layer. FAZ, VD and GCL+ thickness (GCLT) were evaluated using OCTA. Vascular parameters were: FAZ area, perimeter and circularity as well as macular VD (divided by superior, inferior, nasal, temporal e central). GCLT was measured within the 4.5x4.5 mm macular scan area and mean GCLT refers to the average thickness across this entire scanned region, as well as separately for the superior and inferior halves.

OCTA images with poor image quality or significant image artifacts were excluded before the quantitative analysis, including: (1) image quality score less than 40, (2) inaccurate segmentation of tissue layers, (3) motion artifacts (e.g., vessel discontinuity), (4) blurry images, (5) poor centration or (6) signal loss (e.g., due to eye blinking).

**OCT-A imaging processing: FAZ segmentation and quantification.** The area was defined as the size of the segmented FAZ region, and the perimeter was determined by the length of the FAZ contour. Circularity was calculated using the formula $4\pi$ (area/perimeter$^2$), which expresses how closely a shape approximates a perfect circle. Values approaching 1.0 denote higher circularity, whereas lower values indicate greater border irregularity [21]. In ImageJ, the measurement results were shown in pixel units. Because the images for the 4.5 mm × 4.5 mm protocol exported from the OCT were 719 × 719 pixels, the unit of pixel was converted to millimeters at a ratio of 719–4,5.

The Level Sets Macro (LSM) was used as it LSM exhibited greater accuracy and reliability compared to the KSM and inbuilt automated methods [21]. It is a plugin utilizing the theory of partial differential equations that can progressively compare pixel differences with neighboring pixels and converge at the boundary where the differences are the highest (available at https://imagej.net/Level_Sets). After importing the 8-bit grayscale image into ImageJ, inserting an oval to act as an initial seed inside the FAZ is required before running the program (Fig 1A). The shape and the size of the initial seed are not particularly important, but it is essential that the seed be entirely inside the FAZ, preferably located at the center of the FAZ. Based on the optimized parameters, the contour advances and can be viewed in the progress window (Fig 1B). When the contour hits the boundary, segmentation of the FAZ is finished (Fig 1C). Finally, the LSM automatically measures and outputs the FAZ metrics (area, perimeter, and circularity) [21].

The parameters of the LSM were optimized as follows. The Active Contours method was chosen for the LSM rather than Fast Marching because the latter method is prone to leaking, especially when there is a gap at the FAZ boundary. This program advances the contour like a rubber band, with the strength being controlled by the nature of the curvature. The convergence serves as the convergence criterion and compares the changes in contour between two iterations. We used the grayscale of 30 as it performed better compared to other values [21].

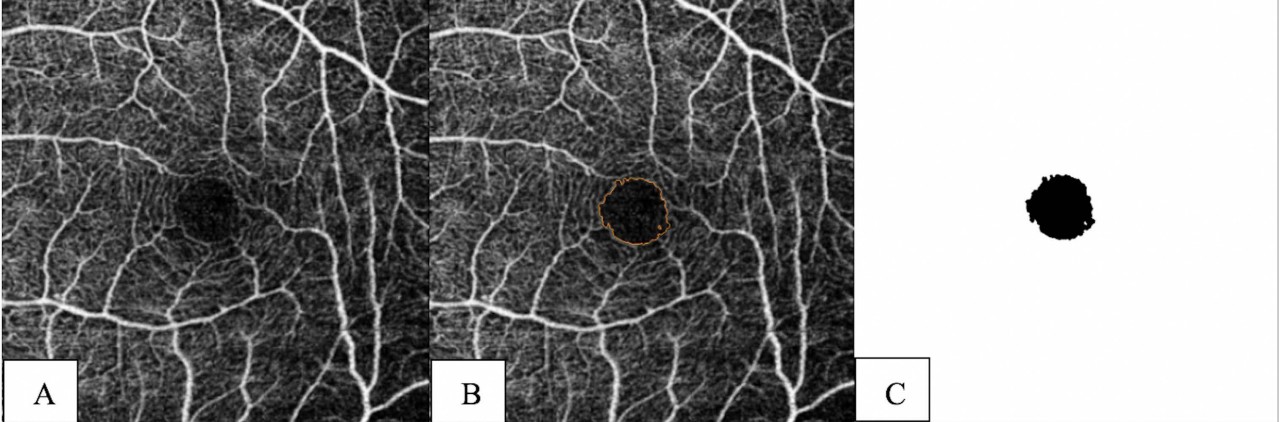

**Fig 1. Segmentation process of the FAZ using Level Sets Macro.** (A) An initial seed at the center of the FAZ is required. (B) Active contour expansion during processing. (C) Final FAZ boundary after segmentation.

The area ($A_{imagej}$) calculated using ImageJ were adjusted for ocular magnification considering axial length of each eye. The adjusted FAZ area ($A_{adjusted}$) (in mm2) was then calculated as follows: $A_{adjusted} = A_{imagej} (ALs/ALm)^2$, where ALs is the axial length of the subject in mm, and ALM is the axial length assumed for the model eye by the manufacturer (24.39 mm) (DRI OCT Triton, Topcon Inc, Tokyo, Japan).

## Visual field assessment

Octopus 900 (Octopus 900, Haag-Streit, AG, Koeniz, Switzerland), white-on-white perimeter was used to evaluate VF of glaucoma patients. G and Macular program were used. The G program has a physiology-based grid of 59 test locations within the central 30 degrees. Locations are clustered more closely together centrally (2.8 degree spacing) with five central foveal locations and 17 test locations in the macular region. Locations are spaced further apart peripherally with emphasis on locations in nasal step regions and with more test locations nasally than temporally. Macular program consists in 45 points one degree spacing in the fovea (central 4 degrees) and 36 points radially oriented from 4 to 10 degrees. Goldmann size III stimuli against a photopic background luminance of 31.4 apostilbs (10 candela/m$^2$), TOP strategy, white-to-white, were used. VF variable analyzed was MD.

## Statistical analysis

Statistical analyses were performed using IBM SPSS (v29.0.2.0) and JAMOVI (v2.4.7.0). Variables were tested for normality using the Shapiro-Wilk test. Among those included in the regression model, only the variables related to the VF were not normally distributed. A regression analysis using mixed linear models was performed to account for the dependence between the two eyes of the same patient. In this model, the patient was included as a random effect, allowing the natural correlation between intra-individual ocular measurements to be considered and enabling more accurate estimation of the fixed effects of interest. Univariate and multivariate analyses were performed. Results are reported as coefficients of determination ($R^2$), p-values, and 95% confidence intervals (CIs). A two-sided alpha level of 0.05 was considered statistically significant.

Additionally, univariable analyses were performed to assess the potential association of gender, intraocular pressure (IOP), and central corneal thickness (CCT) with FAZ parameters. Continuous variables were analyzed using linear mixed-effects models accounting for inter-eye correlation, while gender differences were assessed with independent samples t-tests. Given the available sample size, these variables were not included in the multivariable models to avoid overfitting.

Vascular parameters (FAZ area, perimetry,circularity and also mVD) were considered dependent variables and GCLT independent for structure/structure analysis. For structure/function analysis VF parameters (MD for G and M program) were the dependent variables and vascular parameters, independent. MD values in logarithmic scale were converted to linear scale using the equation [22,23]: $MD(1/Lambert) = 10^{MD(dB)/10}$.

VF sensitivity values are commonly expressed in decibels (dB), which represent a logarithmic scale. Because logarithmic units can distort the magnitude of differences when used in regression analyses, we converted MD values into a linear scale [22]. This transformation provides a proportional representation of retinal sensitivity, allowing more accurate modeling of structure-function relationships in glaucoma [22]. Previous studies have demonstrated that analyzing VF data in linear rather than logarithmic units reduces bias, better reflects biological changes, and improves statistical interpretation of associations with structural and vascular parameters [22,24].

The sample size was calculated with R Software (R 4.4.1, R Foundation for Statistical Computing, Vienna, Austria), for a multiple linear regression model with two independent predictors and an assumed effect size of f2 = 0.20, considering both eyes per patient. An intraclass correlation coefficient (ICC) of 0.65 for FAZ area and a design effect of 1.65 were applied. This adjustment resulted in a required sample size of 86 eyes, corresponding to 43 patients, to achieve 80% power at a significance level of α = 0.05.

## Results

30 women and 27 men were included in this study. The mean age was 66.67 ± 7.49 years old. The mean of MD of M program and G program, GCLT and NFL thickness and FAZ parameters were described in Table 1.

MD of Macular program (mMD) was significantly correlated with FAZ area (p = 0.003) and perimeter (p = 0.004). mMD was not associated with circularity (p = 0.121). For structure/structure analysis, significantly associations were found between FAZ area and central macular VD (p < 0.001; $R^2$ = 0.538) and between FAZ perimeter and central mVD (p < 0.001; $R^2$ = 0.259). FAZ circularity did not show significant correlation with central mVD (p = 0.451). FAZ area was significantly correlated with mean GCLT (p = 0.003; $R^2$ = 0.113) and superior (p < 0.001; $R^2$ = 0.148). FAZ perimeter showed significant correlation with mean GCLT (p = 0.023; $R^2$ = 0.063) and superior GCLT (p = 0.006; $R^2$ = 0.091). FAZ area and FAZ perimeter did not show significant correlation with inferior GCLT (p = 0.054; p = 0.059). FAZ circularity did not show significant correlation with GCLT. In the univariable analysis, IOP and CCT showed no significant associations with any FAZ parameters (p > 0.05). Similarly, independent samples t-tests revealed no significant differences in FAZ parameters between male and female patients (p > 0.05). Considering the sample size, these variables were not incorporated into the multivariable mixed-effects models, which were restricted to age and axial length.

Associations between FAZ and structural/functional parameters were described in Table 2. Fig 2 shows the scatter plot of the relation between mMD and FAZ area. Figs 3–5 show the scatter plots of the relation between FAZ area and mean, superior and inferior GCLT respectively. More advanced stages of glaucoma were shown to have higher FAZ area, higher mMD and lower mean GCLT (Figs 6–8 respectively). Fig 9 exemplifies the integration of OCTA-derived FAZ parameters with high-density macular perimetry and OCT measures in a glaucoma patient.

**Table 1. Clinical characteristics of the study sample.**

| Glaucoma patients | |
|---|---|
| Age | 66.67 ± 7.49 |
| Gender (female patients) | 27 |
| Mean mMD (dB) | 5.59 ± 6.26 |
| Mean MD G program (dB) | 7.91 ± 7.18 |
| Mean central GCLT (µm) | 56.30 ± 6.95 |
| Mean RNFLT (µm) | 75.60 ± 22.90 |
| Mean FAZ area (mm$^2$) | 0.431 ± 0.159 |
| Mean FAZ circularity | 0.387 ± 0.130 |
| Mean FAZ perimeter (mm) | 4,02 ± 1.11 |
| Mean central mVD (%) | 12.9 ± 0.89 |
| Mean central corneal thickness (µm) | 537 ± 29.3 |
| Mean intraocular pressure (mmHg) | 13.9 ± 2.17 |
| Brusini`s GSS (eyes) | 89 |
| Stage 0 | 6 |
| Border | 9 |
| Stage 1 | 10 |
| Stage 2 | 21 |
| Stage 3 | 21 |
| Stage 4 | 12 |
| Stage 5 | 10 |

mMD, mean defect of macular program; MD, mean defect; GCLT, ganglion cell layer thickness; RNFLT: retinal nerve fiber layer thickness; FAZ: foveal avascular zone; mVD: macular vessel density; GSS: glaucoma staging system.

**Table 2. Correlations between FAZ and structural/functional parameters.**

| | FAZ área | | FAZ perimeter | | FAZ circularity | |
|---|---|---|---|---|---|---|
| | R² | p value | R² | P p value | R² | p value |
| **mMD** | 0.079 | 0.023 | 0.151 | 0.006 | 0.035 | 0.121 |
| **mVD** | 0.538 | <0.001 | 0.259 | <0.001 | 0.008 | 0.451 |
| **Mean GCLT** | 0.234 | 0.003 | 0.246 | 0.014 | 0.148 | 0.251 |
| **Superior GCLT** | 0.250 | <0.001 | 0.268 | 0.004 | 0.172 | 0.069 |
| **Inferior GCLT** | 0.068 | 0.054 | 0.219 | 0.059 | 0.137 | 0.684 |

mMD, mean defect of macular program (linear scale); GCLT, ganglion cell layer thickness; FAZ, foveal avascular zone; mVD, macular vessel density.

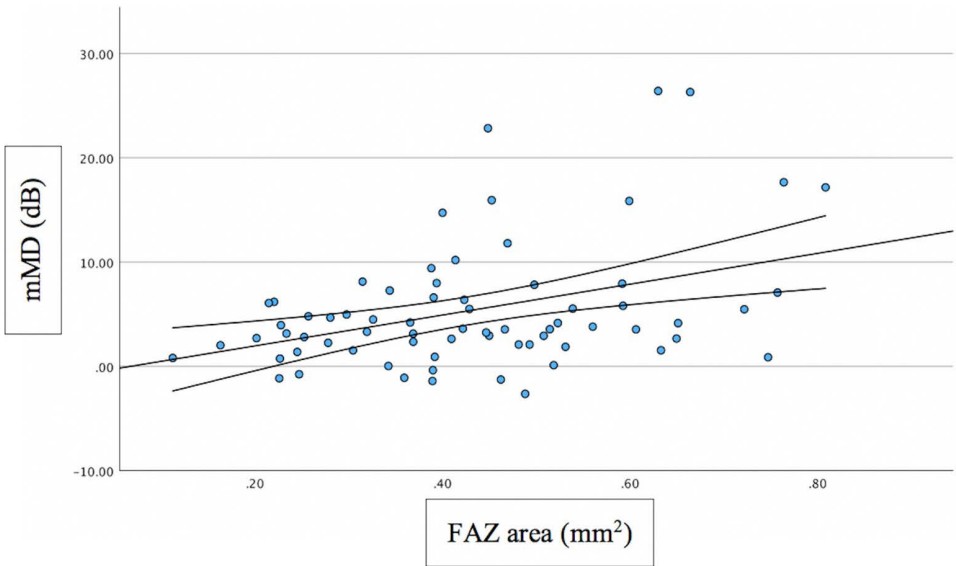

**Fig 2. Scatter plot of the relation between mMD and FAZ area.**

## Discussion

There has been a growing recognition of the role that abnormalities in ocular blood flow and vascular regulatory capacity play in the development and progression of various types of glaucoma [25–28]. Potential indicators of vascular viability include measurements of the FAZ, a unique capillary-free region formed by a ring of interconnected capillaries of foveal vascular plexus [7]. Patients with glaucoma have been shown to have a reduction in mVD [25,26], larger FAZ area and perimeter and loss of circularity [27,28]. Therefore, evaluating FAZ and its correlations may play an important role in glaucoma diagnosis and progression. Our study evaluated the structure and function association of FAZ parameters (area, perimeter and circularity) using macular high density perimetry and OCTA. Our findings highlight the relevance of FAZ metrics as valuable structural indicators in glaucoma. By combining high-density macular perimetry with axial length–adjusted OCTA measurements and multivariate modeling, we demonstrated that FAZ area and perimeter are significantly associated with both macular VF sensitivity and GCLT, particularly in the superior macular region. Notably, FAZ perimeter and circularity were independently associated with age, suggesting that both glaucomatous damage and age-related vascular remodeling contribute to FAZ geometry. Although several associations found in this study were statistically significant, the strength of some correlations, as indicated by R² values, was weak (e.g., R²=0.079 between mMD and FAZ area).

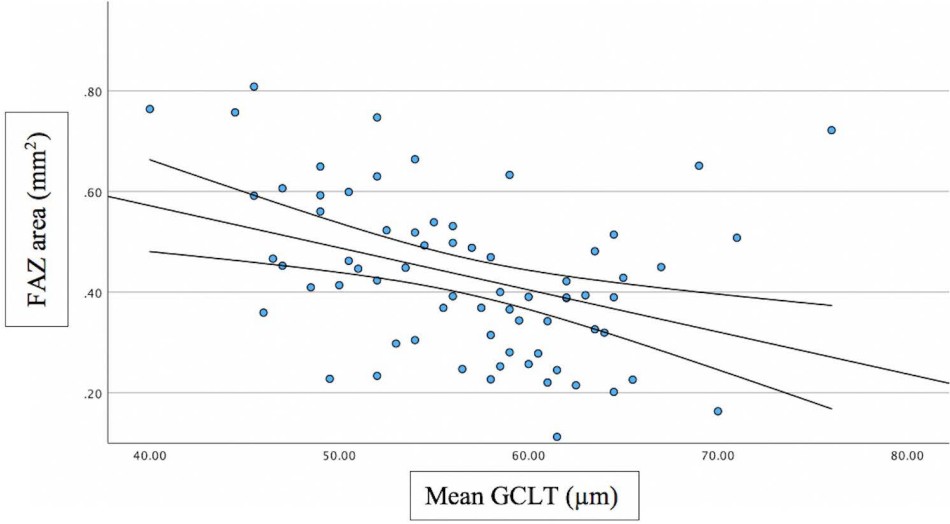

**Fig 3. Scatter plot of the relation between FAZ area and mean GCLT.**

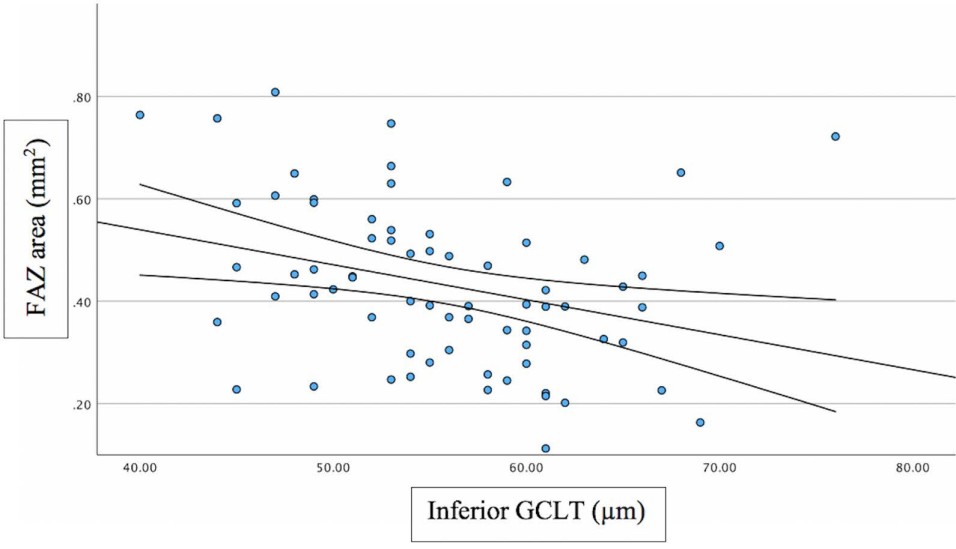

**Fig 4. Scatter plot of the relation between FAZ area and inferior GCLT.**

Previous studies have reported significant relationships between central VF and FAZ parameters [16,29–31]. Consistent with prior reports, our results showed that mMD was significantly correlated with FAZ area (p = 0.023; $R^2$ = 0.079) and perimeter (p = 0.006; $R^2$ = 0.151), though the strength of the association was modest. Higher mMD values were observed in more advanced stages of glaucoma (Fig 7), although Brusini`s glaucoma staging system does not take into account the macula. These findings offer valuable insights into the association between macular microcirculatory disruption and central visual field defects (CVFDs). Although FAZ area and perimeter are relatively unknown parameters for glaucoma evaluation, our findings using OCTA may provide evidence that these parameters can be used as biomarkers for studying the relationship between macular microcirculation and CVFDs in glaucoma and that FAZ measurements via OCTA might

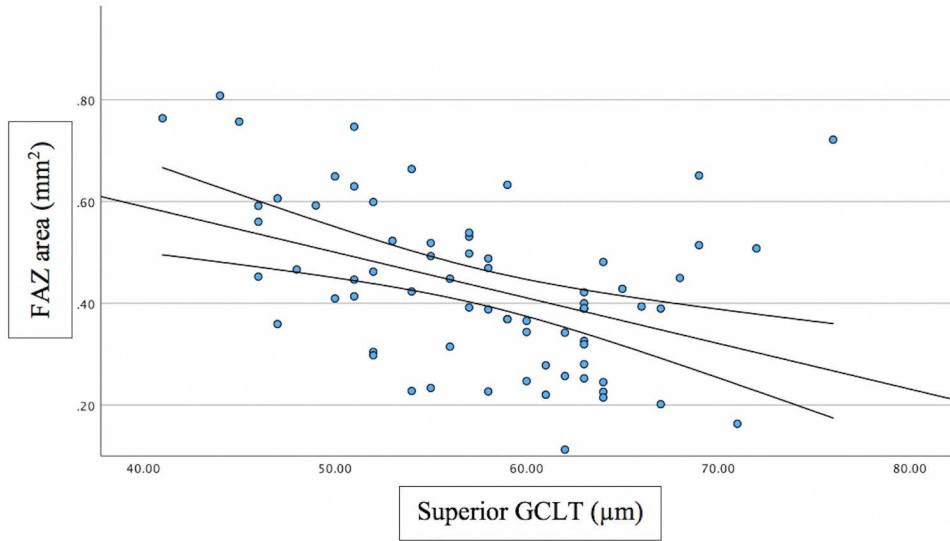

**Fig 5. Scatter plot of the relation between FAZ area and superior GCLT.**

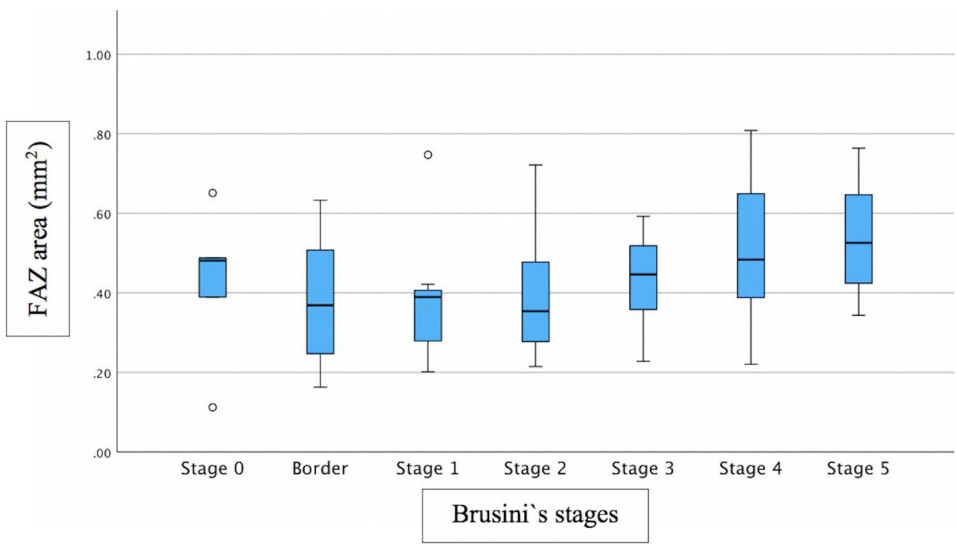

**Fig 6. Box plot of the relation between FAZ area and Brusini`s glaucoma staging system.**

serve as a morphologic index for assessment of eyes with glaucoma. Since the FAZ border is formed by the SVC in the central region of the macula, the involvement of CVF defects may influence the correlation between FAZ parameters and VF mean sensitivity (MS) measurements that are derived globally and regionally [30]. In eyes diagnosed with OAG but without CVF involvement, both the FAZ architecture and CVF MS may remain relatively preserved, which could account for the absence of significant correlations between FAZ parameters and CVF MS [30]. These findings suggest that FAZ area and perimeter may represent valuable structural biomarkers for monitoring functional impairment in glaucoma patients exhibiting CVFDs[31].

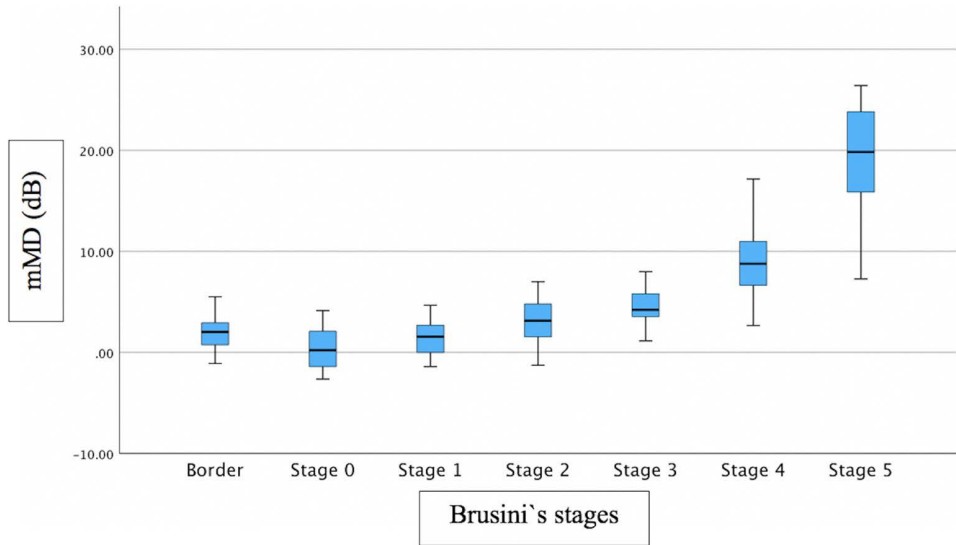

**Fig 7. Box plot of the relation between mMD and Brusini`s glaucoma staging system.**

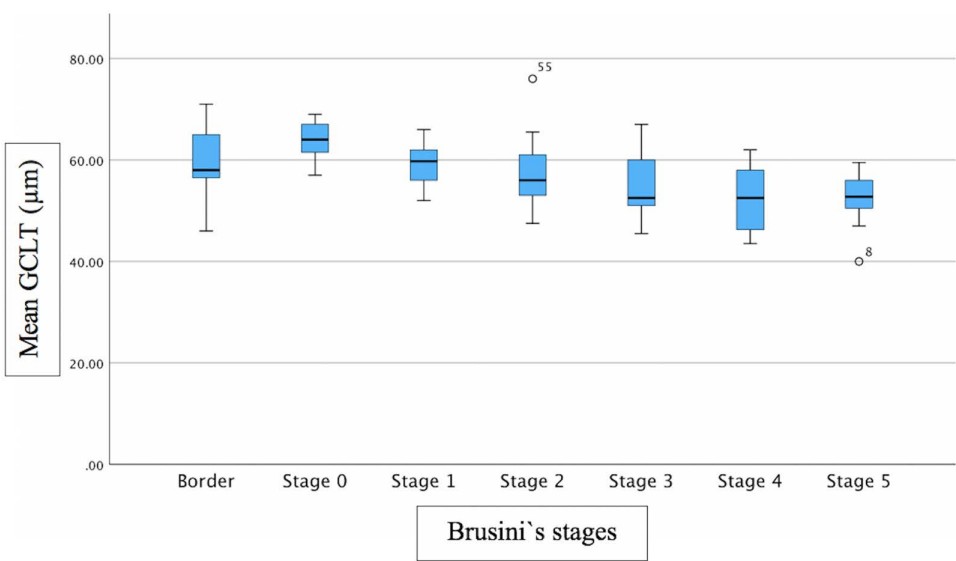

**Fig 8. Box plot of the relation between mean GCLT and Brusini`s glaucoma staging system.**

Previous OCTA studies have consistently shown reduced mVD in glaucoma compared to healthy eyes [13,27,30,32,33]. Moreover, mVD has been reported to correlate negatively with FAZ metrics, with larger FAZ area and perimeter associated with lower mVD and greater disease severity [27,30]. In our study FAZ area and perimeter showed strong and moderate significant correlation with central mVD ($p<0.001$; $R^2=0.538$; $p<0.001$; $R^2=0.259$, respectively). FAZ circularity did not show significant correlation with central mVD.

Previous studies have reported that FAZ area and perimeter are negatively correlated with sctructural parameters, including GCLT and RNFL thickness. These associations were particularly evident with superior GCLT. In contast, FAZ circularity generally showed no significant relationship with GCLT, while several studies consistently confirmed that a

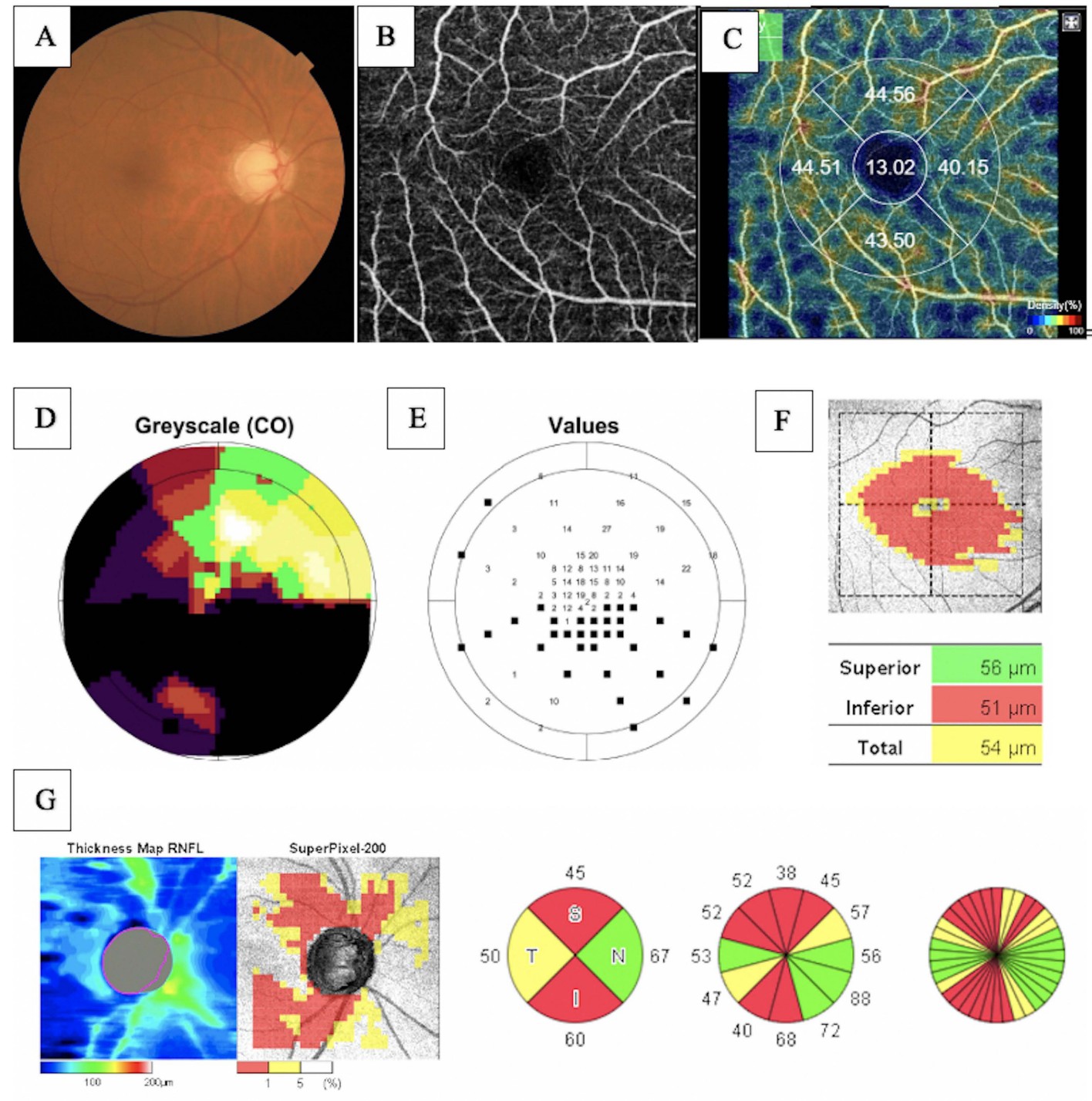

**Fig 9. Representative case of a glaucoma patient illustrating multimodal imaging.** (A) Retinography (B) OCTA – FAZ (C) Macular vessel density (D) Octopus Visual Field Macular Program Greyscale (E) Octopus Visual Field Macular Program threshold sensitivities (F) Ganglion cell layer + inner plexiform layer thickness (GCL+) (G) Peripapillary retinal nerve fiber layer thickness.

larger FAZ area is associated with thinner retinal structural measures. In our study, FAZ area showed moderate significant correlation with mean GCLT ($R^2 = 0.234$; $p = 0.003$) and superior GCLT ($R^2 = 0.250$; $p < 0.001$), but the correlation was not significant between FAZ area and inferior GCLT. FAZ perimeter showed moderate significant correlation with mean GCLT ($R^2 = 0.246$; $p = 0.014$) and superior ($R^2 = 0.268$; $p = 0.004$), but the correlation was not significant between FAZ perimeter and inferior GCLT. FAZ circularity was not significantly correlated with GCLT. The reason that FAZ area and FAZ perimeter showed significant correlation with superior GCLT, but not with inferior GCLT can be explained based on how structural damage occurs as glaucoma progresses. In the early stages of glaucoma, the inferior retina, corresponding to superior hemifield, is involved more frequently [34,35]. In advanced glaucoma, extensive glaucoma damage occurs, with the relative preservation of ganglion cell bodies at superior macula area [36]. According to this, Choi et al [36] found that the relationship with the MD was stronger for the superior RNFL/GCL thickness than for the inferior RNFL/GCL thickness in advanced glaucoma patients.

This rationale is illustrated by representative cases from our cohort (Figs 10 and 11). In Fig 10, which represents an eye with advanced glaucoma, both superior and inferior GCLT were markedly reduced, with extensive macular involvement.

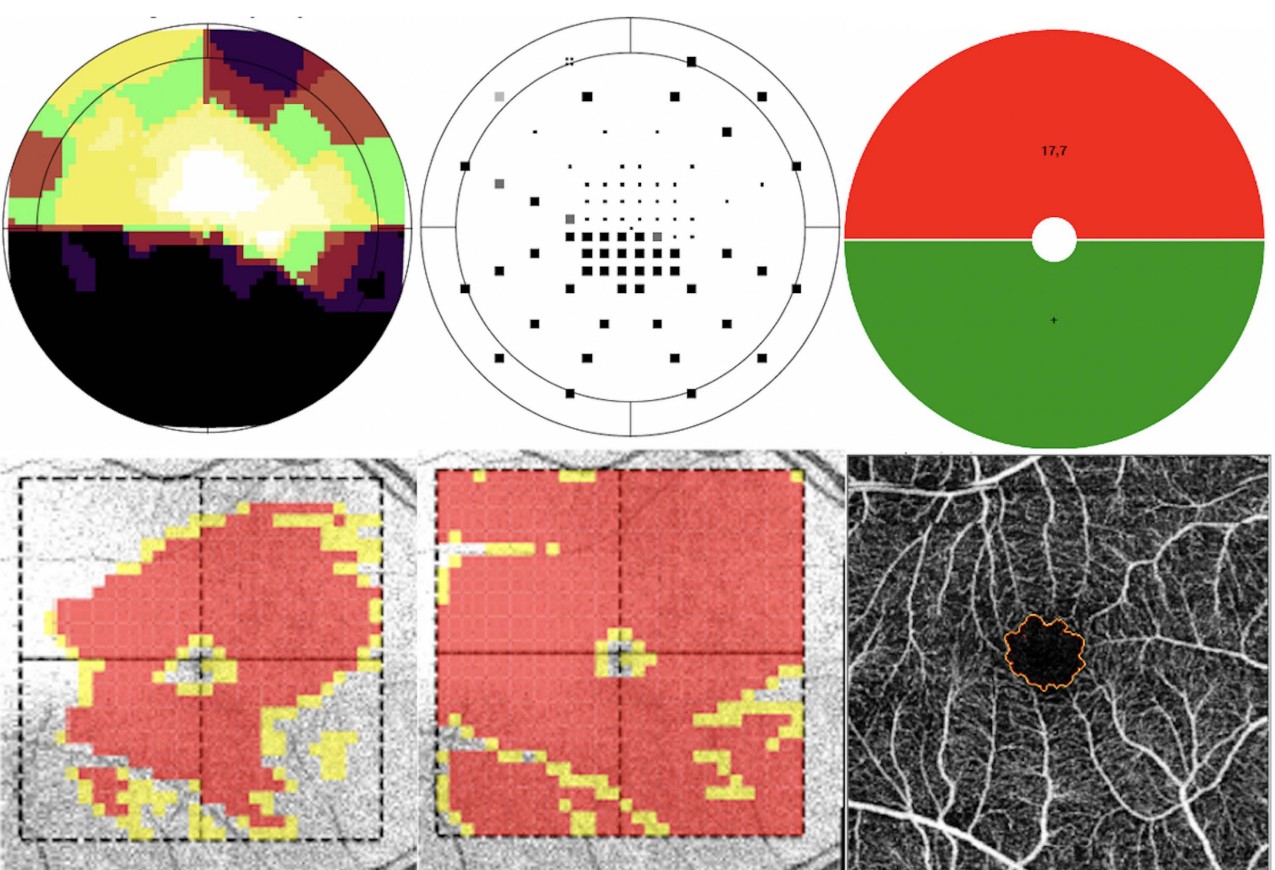

**Fig 10. Representative case with advanced glaucoma.** (A) Octopus Macular Program Greyscale (B) Octopus Macular program corrected probability map (C) Octopus Macular program macula map (D) Ganglion cell layer + inner plexiform layer thickness (GCL+) (E) Retinal nerve fiber layer + ganglion cell layer + inner plexiform layer thickness (GCL++) (F) Optical coherence tomography angiography segmentation of the foveal avascular zone. This case demonstrates advanced glaucoma with extensive macular involvement. Both superior and inferior ganglion cell layer thicknesses are markedly reduced, and visual field testing shows widespread functional loss. The foveal avascular zone is enlarged and irregular, reflecting concomitant microvascular impairment. This example illustrates how advanced disease is associated with structural and vascular alterations that contribute to the stronger correlation of foval avascular zone parameters with superior ganglion cell layer thickness.

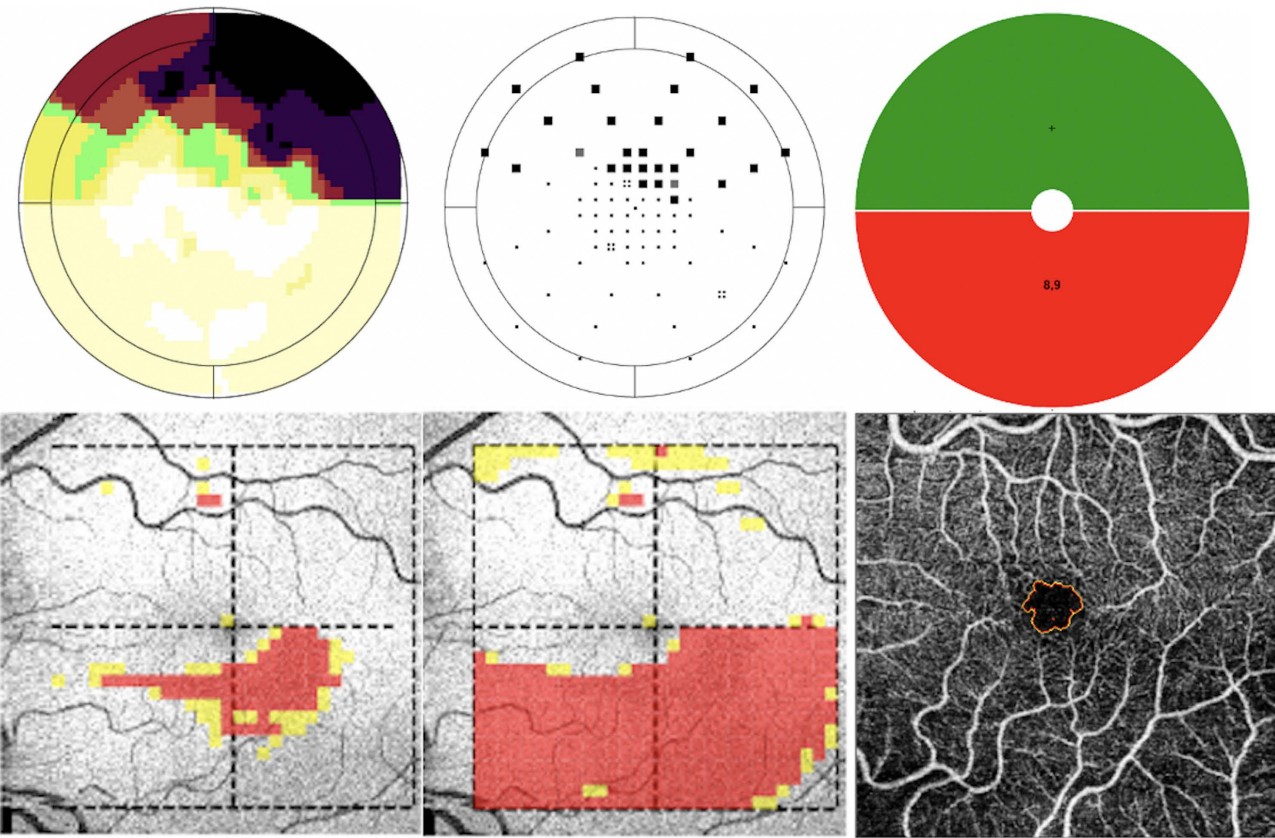

**Fig 11. Representative case with localized inferior macular damage.** (A) Octopus Macular Program Greyscale (B) Octopus Macular Program corrected probability map (C) Octopus Macular program Macula Map (D) Ganglion cell layer + inner plexiform layer thickness (GCL+) (E) Retinal nerve fiber layer + Ganglion cell layer + inner plexiform layer thickness (GCL++) (F) Optical coherence tomography angiography segmentation of the foveal avascular zone. This case shows localized inferior macular damage with corresponding thinning of inferior GCLT. Despite structural loss, FAZ morphology remains preserved, with no evident enlargement or irregularity. This example illustrates that when glaucomatous injury is confined to the inferior macula, vascular changes may not occur, helping explain the absence of significant correlation between FAZ metrics and inferior GCLT.

Consistent with disease severity, the FAZ was enlarged and irregular, reflecting concomitant vascular compromise. In contrast, Fig 11 depicts a case with localized inferior macular damage with a corresponding reduction in inferior GCLT, but the FAZ contour remained preserved without evident vascular alteration. These examples reinforce our findings: in advanced glaucoma, as structural injury becomes widespread, FAZ parameters correlate more strongly with superior GCLT due to relative preservation of ganglion cell bodies in the superior macula. Conversely, in cases with isolated inferior involvement, FAZ changes may not be apparent, explaining the lack of significant association with inferior GCLT. Together, these clinical cases provide biological plausibility to the stronger correlations observed with superior rather than inferior GCLT in our analysis.

In our study, FAZ circularity consistently showed no correlation with structural and functional parameters, despite being hypothesized as meaningful. This can be explained by several factors. First, circularity is a shape-based metric that reflects irregularity of the FAZ border, which appears to be more strongly influenced by physiological aging than by glaucomatous damage [37]. Indeed, we observed that circularity was consistently and inversely associated with age, but not with GCLT or macular sensitivity, in line with previous reports demonstrating that age-related microvascular remodeling increases FAZ irregularity independently of disease status [37–39]. Second, unlike FAZ area, circularity

does not change monotonically with expansion of the FAZ: small local irregularities or segmentation artifacts can substantially affect the index, thereby increasing variability and weakening correlations [39,40]. Finally, earlier studies have also reported absent or weak associations between circularity and glaucoma severity [16,29], while FAZ area and perimeter showed stronger relationships with both structural loss and mVD [13,25,30]. Therefore, our findings reinforce the notion that FAZ circularity may primarily capture age-dependent vascular irregularities, while FAZ area and perimeter reflect glaucomatous structural and functional loss. In our study, many of the associations demonstrated relatively low coefficients of determination ($R^2 < 0.25$). This finding is not unexpected in glaucoma research [41]. Structure-function relationships in glaucoma are inherently complex and influenced by multiple biological and methodological factors, including variability in VF testing, inter-individual anatomical differences, and age-related changes [42–47]. Low $R^2$ values therefore do not necessarily indicate that the associations lack clinical significance. Rather, they suggest that FAZ metrics should not be interpreted in isolation but in conjunction with other structural and functional parameters [48]. Importantly, even modest correlations may provide incremental diagnostic or prognostic value when integrated into multimodal assessment, particularly for longitudinal monitoring where small but consistent changes may indicate disease progression [49,50]. Thus, while our results do not justify clinical implementation of FAZ parameters as stand-alone biomarkers, they reinforce their potential role as complementary metrics within a broader multimodal glaucoma evaluation framework.

FAZ area has typically been manually measured and only some OCTA machines are equipped with automated measurement software [21]. The manual measurement method is limited as it is subjective in determining how many end points should be plotted or which points should be selected as end points. Moreover, low reliability and repeatability of automated measurement have been reported [21,51]. In this study, we used an objective way to measure FAZ parameters [21]. Image J (National Institutes of Health, Bethesda, MD) was used to automatically measure FAZ area, perimeter and circularity. The Level Sets Macro (LSM) was used as LSM exhibited greater accuracy and reliability compared to the KSM and inbuilt automated methods [21]. LSM automatically measured and output the FAZ metrics (area, perimeter, and circularity). Then, FAZ area was adjusted for axial length (AL) [23]. Although FAZ area is influenced by AL, few studies have considered the magnification effect. Linderman et al [23] reported that if this magnification effect is not corrected, measurements of the FAZ will be inaccurate up to 31%. Without axial length correction, the FAZ area is overestimated in the eye with a shorter axial length and underestimated in the eye with a longer axis [52]. As FAZ circularity and perimeter is not impacted by ocular magnification, we did not adjust it by AL [40].

Our approach included both univariate and multivariate mixed-effects models, which accounted for inter-eye correlation and allowed us to isolate the influence of age and GCLT on vascular parameters. Univariate analyses revealed significant associations between age and FAZ perimeter and circularity ($p < 0.001$ for both), indicating that aging is accompanied by increased irregularity and expansion of the FAZ border. No significant association between age and FAZ area or mVD was detected in univariate models, suggesting that age may influence FAZ shape more than size. These findings align with the hypothesis that age-related microvascular remodeling alters FAZ geometry, independent of glaucomatous damage. Our multivariate linear mixed-effects models, which accounted for inter-eye correlations, provided additional insights into the factors influencing FAZ parameters. When evaluating FAZ area as the dependent variable, age did not show a statistically significant association after adjusting for GCLT, whether using mean, superior, or inferior GCLT values ($p = 0.233$, $p = 0.133$, and $p = 0.336$, respectively).

The age-related increase in FAZ perimeter and decrease in circularity likely reflect microvascular remodeling and capillary dropout associated with normal aging [37]. Previous studies suggest that perifoveal capillaries become sparser and more irregular with advancing age, contributing to loss of vascular symmetry [37–39]. These changes may overlap with glaucomatous damage, complicating disease staging. Our findings indicate that careful consideration of age-related vascular alterations is essential when interpreting FAZ metrics, as they may act as confounding or additive factors in glaucoma evaluation.

Interestingly, FAZ perimeter showed a consistent and significant positive association with age in all models (p < 0.001), even after controlling for GCLT. This suggests that, beyond glaucomatous changes, age may independently contribute to the expansion of the FAZ perimeter, possibly through age-related microvascular alterations or structural remodeling of the perifoveal capillary network. Moreover, FAZ circularity exhibited a distinct pattern: it was not significantly associated with any GCLT parameter but demonstrated a strong and consistent inverse relationship with age across all tested models (p < 0.001). The negative coefficient for age indicates a progressive reduction in circularity with aging, reflecting increased FAZ irregularity that appears to be age-dependent rather than disease-specific.

Taken together, these findings suggest a differential influence of age on FAZ geometry. While FAZ area seems more reflective of glaucomatous structural loss, perimeter and particularly circularity may be more affected by physiological aging. These distinctions are clinically relevant and should be considered when interpreting OCTA-derived FAZ metrics in the evaluation of glaucoma patients.

With respect to mVD, we observed significant correlations with FAZ area and mMD, but no association with age in either univariate or multivariate models. These findings suggest that mVD may serve as a more specific marker of glaucomatous vascular compromise rather than of age-related change. Importantly, no significant interaction was found between age and GCLT in relation to any FAZ parameter or mVD, indicating that the effects of aging and neurodegeneration on macular microvasculature appear to be additive rather than interactive.

Our results complement and expand previous literature by providing a more comprehensive analysis of the relationship between FAZ parameters and central VF sensitivity. Igarashi et al.[31] reported an association between central VF sensitivity and FAZ area using the HFA 10−2 protocol; however, they did not evaluate FAZ perimeter or circularity. Kwon et al.[16,30] explored the relationship between VF sensitivity and multiple FAZ metrics—area, perimeter, and circularity—using SAP (HFA 24−2) but did not account for axial length magnification nor employ multivariate models to adjust for potential structural confounders. In contrast, our study implemented a more rigorous methodological approach, incorporating axial length-corrected FAZ measurements and multivariate analyses adjusted for GCLT. Similarly, Li et al. [53] identified associations between VF sensitivity and FAZ area and circularity using the HFA 24−2 program. Nishida et al. [29] also reported significant relationships between FAZ metrics (area and circularity) and VF sensitivity assessed with both the HFA 24−2 and HFA 10−2 protocols. A summary of the FAZ parameters analyzed, perimetric protocols applied, and whether adjustments for age and axial length were included in each study is provided in Table 3. A crucial methodological distinction is that, unlike previous studies, we used high-density macular perimetry, which provides a finer sampling of the central VF and improves spatial correspondence with anatomical structures captured by OCT and OCTA. This approach enhances the sensitivity to detect localized functional losses and their vascular correlates, strengthening structure-function correlations in the macular region.

To our knowledge, this is one of the most comprehensive evaluations to date of FAZ morphology in glaucoma incorporating multivariate models, age effects, high-density perimetry, and axial-length–corrected OCTA measurements. The finding that age independently influences FAZ perimeter and circularity, but not area, while FAZ area is predominantly

**Table 3. Summary of FAZ studies.**

| Author (Year) | Analysis Type | FAZ Parameter | VF Program | Age Adjustment | Axial Length Adjustment |
|---|---|---|---|---|---|
| Igarashi et al. (2020) | Structural and Functional | Area | HFA 10−2 | Yes | Not mentioned |
| Kwon et al. (2018) | Structural and Functional | Area, Perimeter | HFA 24−2 | Yes | Not mentioned |
| Kwon et al. (2017) | Structural and Functional | Area, Circularity | HFA 24−2 | Yes | Not mentioned |
| Li et al. (2022) | Structural and Functional | Area, Circularity | HFA 24−2 | Yes | Yes |
| Nishida et al. (2023) | Structural and functional | Area, Circularity | HFA 24−2 and 10−2 | Yes | Yes |

FAZ, foveal avascular zone; VF, visual field; HFA, Humphrey field analyzer.

linked to glaucomatous structural loss, provides important clinical insight. These results underscore the need to consider both vascular and structural metrics—along with patient age—when interpreting OCTA findings in glaucoma. Furthermore, our study supports the clinical utility of FAZ circularity and perimeter as complementary biomarkers that may detect early age-related microvascular irregularities, while FAZ area remains more indicative of disease progression. Recognizing the differential contributions of age and glaucomatous damage to FAZ geometry may aid in more accurate diagnosis, staging, and longitudinal monitoring of glaucoma patients. Evaluating the strength of structure–function associations in glaucoma through various structural and functional assessment methods has important clinical relevance. It enhances the understanding of disease progression and supports the selection of appropriate diagnostic tools for patient follow-up [49].

This study has several limitations. First, only the superficial FAZ area was investigated, as deep FAZ area has lower reproducibility compared with superficial FAZ area [54]. Moreover, contrast sensitivity depends upon the size of the test stimulus, in other words, it is determined by spatial summation [55,56]. Standard GIII stimulus, being larger than the Ricco area, exceeds the total spatial summation area and it may not demonstrate glaucomatous damage satisfactorily [20,55,56]. Smaller stimuli or microperimetry, which use precise retinal tracking and allow denser sampling with size I or II stimuli, could potentially improve the sensitivity to early macular damage and yield stronger structure function correlations [15,19]. Future studies comparing high-density perimetry with microperimetry or using smaller Goldmann stimuli may clarify the incremental diagnostic value of these approaches. Finally, this study was restricted to glaucomatous eyes, therefore generalizability of our results to eyes with other pathologies may be limited.

This study provides robust evidence supporting the role of FAZ metrics as relevant structural biomarkers in glaucoma. By integrating high-density macular perimetry with axial length–corrected OCTA parameters and multivariate modeling, we demonstrated that FAZ area and perimeter are significantly associated with both macular VF sensitivity and GCLT, particularly in the superior macular region. Moreover, we found that FAZ perimeter and circularity are independently influenced by age, suggesting a dual contribution of glaucomatous damage and age-related microvascular remodeling to FAZ geometry. The observed structure-structure and structure-function correlations underscore the clinical utility of FAZ-based analysis in capturing microvascular alterations related to glaucomatous progression. These findings reinforce the potential of FAZ area, perimeter, and circularity as complementary, non-invasive biomarkers for individualized diagnosis, staging, and monitoring of glaucoma.

## Author contributions

**Conceptualization:** Gustavo Coelho Caiado, Sergio Henrique Teixeira, Tiago Santos Prata, Carolina Pelegrini Barbosa Gracitelli, Augusto Paranhos Jr.

**Data curation:** Gustavo Coelho Caiado, Gustavo Albrecht Samico, Gilvan Vilarinho da Silva Filho.

**Formal analysis:** Gustavo Coelho Caiado, Gustavo Albrecht Samico, Gilvan Vilarinho da Silva Filho, Sergio Henrique Teixeira, Tiago Santos Prata, Carolina Pelegrini Barbosa Gracitelli, Augusto Paranhos Jr.

**Investigation:** Gustavo Coelho Caiado, Gustavo Albrecht Samico, Gilvan Vilarinho da Silva Filho, Augusto Paranhos Jr.

**Methodology:** Gustavo Coelho Caiado, Augusto Paranhos Jr.

**Project administration:** Gustavo Coelho Caiado, Sergio Henrique Teixeira, Carolina Pelegrini Barbosa Gracitelli, Augusto Paranhos Jr.

**Resources:** Gustavo Coelho Caiado.

**Software:** Gustavo Coelho Caiado.

**Supervision:** Gustavo Coelho Caiado, Sergio Henrique Teixeira, Tiago Santos Prata, Carolina Pelegrini Barbosa Gracitelli, Augusto Paranhos Jr.

**Validation:** Gustavo Coelho Caiado, Gustavo Albrecht Samico, Gilvan Vilarinho da Silva Filho, Sergio Henrique Teixeira, Tiago Santos Prata, Carolina Pelegrini Barbosa Gracitelli, Augusto Paranhos Jr.

**Visualization:** Gustavo Coelho Caiado, Sergio Henrique Teixeira, Augusto Paranhos Jr.

**Writing – original draft:** Gustavo Coelho Caiado.

**Writing – review & editing:** Gustavo Coelho Caiado, Carolina Pelegrini Barbosa Gracitelli, Augusto Paranhos Jr.

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
