## [Decision Letter · Decision Letter 0]

16 Aug 2025

PONE-D-25-37909High-density perimetry in the assessment of foveal avascular zone and macular structure in glaucomaPLOS ONE

Dear Dr. Paranhos Jr,

Thank you for submitting your manuscript to PLOS ONE. After careful consideration, we feel that it has merit but does not fully meet PLOS ONE’s publication criteria as it currently stands. Therefore, we invite you to submit a revised version of the manuscript that addresses the points raised during the review process.

We look forward to receiving your revised manuscript.

Kind regards,

Shinji Kakihara, M.D.,Ph.D.

Academic Editor

PLOS ONE

Journal Requirements:

https://journals.lww.com/ijo/Fulltext/2022/12000/Analysis_of_aerobic_exercise_influence_on.29.aspx

https://doi.org/10.1371/journal.pone.0235571

In your revision ensure you cite all your sources (including your own works), and quote or rephrase any duplicated text outside the methods section. Further consideration is dependent on these concerns being addressed.

3. In the online submission form, you indicated that your data will be submitted to a repository upon acceptance. We strongly recommend all authors deposit their data before acceptance, as the process can be lengthy and hold up publication timelines. Please note that, though access restrictions are acceptable now, your entire minimal dataset will need to be made freely accessible if your manuscript is accepted for publication. This policy applies to all data except where public deposition would breach compliance with the protocol approved by your research ethics board. If you are unable to adhere to our open data policy, please kindly revise your statement to explain your reasoning and we will seek the editor's input on an exemption.

4. Please ensure that you refer to Figure 9 in your text as, if accepted, production will need this reference to link the reader to the figure.

Additional Editor Comments:

Your manuscript has been reviewed by two experts in the field. The reviewers have provided constructive suggestions to further strengthen its rigor and clarity. Please respond to each comment point-by-point and revise the manuscript accordingly, addressing them as fully as possible. Thank you.

Reviewers' comments:

Reviewer's Responses to Questions

**Comments to the Author**

1. Is the manuscript technically sound, and do the data support the conclusions?

Reviewer #1: Yes

Reviewer #2: Yes

2. Has the statistical analysis been performed appropriately and rigorously? 

Reviewer #1: Yes

Reviewer #2: Yes

3. Have the authors made all data underlying the findings in their manuscript fully available?

Reviewer #1: Yes

Reviewer #2: Yes

4. Is the manuscript presented in an intelligible fashion and written in standard English?

Reviewer #1: Yes

Reviewer #2: Yes

5. Review Comments to the Author

Reviewer #1: 1. Consider adding the mean age and glaucoma severity distribution to better contextualize the findings in the abstract.

2. Clearly state in the abstract that FAZ perimeter and circularity are mainly age-related, while area is linked to glaucomatous damage.

3. In the methods, the sample size calculation assumes independence between eyes; however, the authors later account for inter-eye correlation using mixed models. This should be clarified.

4. Clarify the rationale for converting MD to linear scale using the formula (10^MD/10). Not all readers will be familiar with this.

5. The strength of correlations (R² < 0.25 in most cases) is low. Consider discussing this in more detail—do these justify clinical implementation?

6. It is unclear why FAZ circularity consistently showed weak or no correlation despite being hypothesized as meaningful. This deserves more explanation.

7. The discussion could be more concise. The paper references multiple prior studies in great detail but should focus more on how this study's results compare, contrast, or advance the field.

8. The explanation for the stronger correlation with superior GCLT is plausible but not deeply convincing without visual field maps or regional segmentation figures.

9. Consistently use either macular vessel density (mVD) or just vessel density. Avoid switching terms unnecessarily.

10. Specify if “mean GCLT” is across the whole macula or just the 4.5x4.5 mm central area.

11. The study mentions using Goldmann III stimulus, which may not detect subtle macular defects. Consider discussing whether using microperimetry

12. or smaller stimuli could yield better structure-function correlation.

13. There are some grammatical inconsistencies (e.g., “perimetry and circularity and also vessel density”).

14. Check and standardize the use of abbreviations. For example, “FAZ,” “GCLT,” and “MD” are sometimes repeated in full.

Reviewer #2: The authors designed a study and assessed the relationship between FAZ morphology, macular visual field sensitivity, vessel density, and ganglion cell layer thickness using OCTA and high-density macular perimetry. Automated, axial length–corrected FAZ measurements were analyzed with mixed-effects models accounting for age and inter-eye correlation. FAZ area and perimeter were significantly associated with macular VF defect, mVD, and GCLT, with stronger correlations in the superior macula. FAZ perimeter and circularity were independently related to age, supporting their role as complementary biomarkers reflecting both glaucomatous damage and age-related vascular changes. The study is interesting and well written. Here are some comments that I believe will help improve the study.

1. In the methods, you mentioned SE within +3 and -6 were excluded. Please clarify if this correct or you meant anything beyond that was excluded since between +3 and -6 is a formal inclusion criteria.

2. As I understood, in your multivariable mixed effect model, you considered age and axial length. Based on important landmark studies, it is crucial that you include age, gender, IOP and CCT as well. Please include those variables and report the results of that model.

3. Figures 6-8 are box plots and please correct the figure legend (you incorrectly mentioned scatter plot)

4. The discussion is comprehensive but at times overly long, with some redundancy between background context, literature comparisons, and study findings. Consider streamlining the text to focus on the most novel contributions of this study—particularly the combination of high-density macular perimetry, axial length–corrected OCTA, and multivariate modeling—while moving more routine literature summaries to the introduction. This would make the discussion more concise and highlight the unique value of your work.

5. Several reported correlations (e.g., FAZ area vs. mMD) are statistically significant but have low R² values. While the manuscript acknowledges this, the clinical implications of such modest effect sizes could be further clarified. It may help to discuss whether these associations, despite being weak, would be meaningful for clinical decision-making or longitudinal monitoring.

6. The comparisons with previous studies are detailed, but in some cases they read as a list of findings rather than a synthesis. You might consider summarizing the key differences between your study and prior work in a table (e.g., study population, imaging protocol, AL correction, statistical approach) to make these distinctions more immediately clear to the reader.

7. The finding that age independently affects FAZ perimeter and circularity is important, but the discussion could benefit from more elaboration on the biological mechanisms behind these changes and whether they may confound glaucoma staging. This would strengthen the translational value of the results.

6. PLOS authors have the option to publish the peer review history of their article (what does this mean? ). If published, this will include your full peer review and any attached files.

**Do you want your identity to be public for this peer review?** For information about this choice, including consent withdrawal, please see our Privacy Policy .

Reviewer #1: No

Reviewer #2: No

---

## [Author Response · Author response to Decision Letter 1]

29 Aug 2025

Dear Prof. Dr. Shinji Kakihara and reviewers

Enclose please find our reviewed manuscript entitled “High-density perimetry in the assessment of foveal avascular zone and macular structure in glaucoma”, to be considered for publication as a Research Article in PLOS ONE. Please, see our response to reviewers in following pages.

Please contact me with any questions. Look forward for your reply.

Sincerely,

Augusto Paranhos Jr, MD, PhD

Professor from Federal University of São Paulo, São Paulo, Brazil

augusto.paranhos@gmail.com

Telephone: (+55 11) 5085 2010

Fax number: (+55 11) 5085 2000

Journal requeirements

Comment 1: Please ensure that your manuscript meets PLOS ONE's style requirements, including those for file naming.

Response: We thank the editorial office for this reminder. The revised manuscript has been carefully formatted to meet the PLOS ONE style requirements, and the file names were adjusted according to the journal’s guidelines.

Comment 2: We noticed you have some minor occurrence of overlapping text with the following previous publication(s).

Response: We revised the manuscript thoroughly to eliminate overlapping text with the cited publications. In the revised version, all relevant sources have been appropriately cited, and duplicated text has been rephrased to ensure originality and compliance with the journal’s policy.

Comment 3: In the online submission form, you indicated that your data will be submitted to a repository upon acceptance.

Response: We appreciate this guidance. All the data are available in the Zenodo repository at: https://doi.org/10.5281/zenodo.17000133

Comment 4: Please ensure that you refer to Figure 9 in your text.

Response: We have revised the manuscript to explicitly refer to Figure 9 in the Results section, ensuring that all figures are properly cited in the text.

Comment 5: If the reviewer comments include a recommendation to cite specific previously published works, please review and evaluate these publications.

Response: We carefully evaluated all recommended works. Relevant publications have been added and cited in the revised version, while others deemed not directly relevant were not cited, in accordance with the journal’s instructions.

Reviewer 1

Comment 1: Consider adding the mean age and glaucoma severity distribution to better contextualize the findings in the abstract.

Response: Thank you. We added the mean age and glaucoma severity distribution to the abstract to improve contextualization of the study population.

Comment 2: Clearly state in the abstract that FAZ perimeter and circularity are mainly age-related, while area is linked to glaucomatous damage.

Response: Thank you. The abstract was revised to explicitly state that FAZ perimeter and circularity are primarily age-related, while FAZ area is linked to glaucomatous damage.

Comment 3: In the methods, the sample size calculation assumes independence between eyes; however, the authors later account for inter-eye correlation using mixed models. This should be clarified.

Response: We thank the reviewer for this important observation. In the original version, the description of the sample size calculation did not explicitly account for the correlation between both eyes of the same patient, which could give the impression of independence. To address this, we revised the Methods section to clarify that the calculation was based on a multiple linear regression model considering both eyes per patient. Specifically, we applied an intraclass correlation coefficient (ICC) of 0.65 for FAZ area and a design effect of 1.65 to adjust for inter-eye correlation. This adjustment resulted in a required sample size of 86 eyes (43 patients) to achieve 80% power at a significance level of a =0.05. The revised text now makes clear that the inter-eye dependence was incorporated into the sample size estimation, ensuring methodological consistency with the mixed-effects models used in the analysis.

Comment 4: Clarify the rationale for converting MD to linear scale using the formula (10^MD/10). Not all readers will be familiar with this.

Response: Thank you. We added an explanation in the Methods to justify the conversion of MD to a linear scale, for clarity: “VF sensitivity values are commonly expressed in decibels (dB), which represent a logarithmic scale. Because logarithmic units can distort the magnitude of differences when used in regression analyses, we converted MD values into a linear scale[24]. This transformation provides a proportional representation of retinal sensitivity, allowing more accurate modeling of structure-function relationships in glaucoma[24]. Previous studies have demonstrated that analyzing VF data in linear rather than logarithmic units reduces bias, better reflects biological changes, and improves statistical interpretation of associations with structural and vascular parameters[24,25].”

Comment 5: The strength of correlations (R² < 0.25 in most cases) is low. Consider discussing this in more detail—do these justify clinical implementation?

Response: Thank you. We expanded the Discussion to address the correlations, contextualizing them within glaucoma literature and discussing their translational implications: “In our study, many of the associations demonstrated relatively low coefficients of determination (R²< 0.25). This finding is not unexpected in glaucoma research[42]. Structure-function relationships in glaucoma are inherently complex and influenced by multiple biological and methodological factors, including variability in VF testing, inter-individual anatomical differences, and age-related changes[43–48]. Low R2 values therefore do not necessarily indicate that the associations lack clinical significance. Rather, they suggest that FAZ metrics should not be interpreted in isolation but in conjunction with other structural and functional parameters[49]. Importantly, even modest correlations may provide incremental diagnostic or prognostic value when integrated into multimodal assessment, particularly for longitudinal monitoring where small but consistent changes may indicate disease progression[50,51]. Thus, while our results do not justify clinical implementation of FAZ parameters as stand-alone biomarkers, they reinforce their potential role as complementary metrics within a broader multimodal glaucoma evaluation framework.”

Comment 6: It is unclear why FAZ circularity consistently showed weak or no correlation despite being hypothesized as meaningful. This deserves more explanation.

Response: Thank you. We added more detailed discussion of potential biological and methodological reasons for the weak associations of FAZ circularity, supported by relevant references: “In our study, FAZ circularity consistently showed no correlation with structural and functional parameters, despite being hypothesized as meaningful. This can be explained by several factors. First, circularity is a shape-based metric that reflects irregularity of the FAZ border, which appears to be more strongly influenced by physiological aging than by glaucomatous damage[38]. Indeed, we observed that circularity was consistently and inversely associated with age, but not with ganglion cell layer thickness or macular sensitivity, in line with previous reports demonstrating that age-related microvascular remodeling increases FAZ irregularity independently of disease status[38–40]. Second, unlike FAZ area, circularity does not change monotonically with expansion of the FAZ: small local irregularities or segmentation artifacts can substantially affect the index, thereby increasing variability and weakening correlations[40,41]. Finally, earlier studies have also reported absent or weak associations between circularity and glaucoma severity[16,30], while FAZ area and perimeter showed stronger relationships with both structural loss and mVD[13,26,31]. Therefore, our findings reinforce the notion that FAZ circularity may primarily capture age-dependent vascular irregularities, while FAZ area and perimeter reflect glaucomatous structural and functional loss.”

Comment 7: The discussion could be more concise. The paper references multiple prior studies in great detail but should focus more on how this study's results compare, contrast, or advance the field.

Response: Thank you. The Discussion was streamlined, with reduced redundancy and greater focus on the novel contributions of our findings.

Comment 8: The explanation for the stronger correlation with superior GCLT is plausible but not deeply convincing without visual field maps or regional segmentation figures.

Response: We thank the reviewer for this valuable suggestion. As recommended, we added representative cases with corresponding visual field maps and regional structural/vascular segmentation to illustrate the rationale for the stronger correlation with superior GCLT. These new figures (Fig 10 and Fig 11) depict complementary scenarios: one eye with advanced glaucoma showing diffuse GCLT loss, macular involvement, and enlarged/irregular FAZ (Fig 10), and another eye with localized inferior macular damage and preserved FAZ morphology (Fig 11). The inclusion of these examples strengthens the biological plausibility of our findings, as they demonstrate how FAZ alterations are more apparent in advanced disease with superior macular preservation, but may remain absent when glaucomatous injury is restricted to the inferior macula. We believe that these visual field maps and regional segmentations provide the additional evidence requested by the reviewer, clarifying and supporting our interpretation.

Comment 9: Consistently use either macular vessel density (mVD) or just vessel density. Avoid switching terms unnecessarily.

Response: Thank you. Terminology was standardized throughout the text, consistently using “macular vessel density (mVD).”

Comment 10: Specify if “mean GCLT” is across the whole macula or just the 4.5x4.5 mm central area.

Response: Thank you. We clarified that “mean GCLT” refers to the 4.5×4.5 mm central macular scan area: “GCLT was measured within the 4.5x4.5 mm macular scan area and mean GCLT refers to the average thickness across this entire scanned region, as well as separately for the superior and inferior halves.”

Comment 11: The study mentions using Goldmann III stimulus, which may not detect subtle macular defects. Consider discussing whether using microperimetry or smaller stimuli could yield better structure-function correlation.

Response: Thank you. We added this point to the Discussion, noting that microperimetry or smaller stimuli might provide greater sensitivity for subtle macular defects: “Moreover, contrast sensitivity depends upon the size of the test stimulus, in other words, it is determined by spatial summation[57,58]. Standard GIII stimulus, being larger than the Ricco area, exceeds the total spatial summation area and it may not demonstrate glaucomatous damage satisfactorily[20,57,58]. Smaller stimuli or microperimetry, which use precise retinal tracking and allow denser sampling with size I or Il stimuli, could potentially improve the sensitivity to early macular damage and yield stronger structure function correlations[15,19]. Future studies comparing high-density perimetry with microperimetry or using smaller Goldmann stimuli may clarify the incremental diagnostic value of these approaches.”

Comment 12: There are some grammatical inconsistencies (e.g., “perimetry and circularity and also vessel density”).

Response: Thank you. All inconsistencies were corrected.

Comment 13: Check and standardize the use of abbreviations. For example, “FAZ,” “GCLT,” and “MD” are sometimes repeated in full.

Response: Thank you. Abbreviations were reviewed and standardized throughout the manuscript.

Reviewer 2

Comment 1: In the methods, you mentioned SE within +3 and -6 were excluded. Please clarify if this correct or you meant anything beyond that was excluded since between +3 and -6 is a formal inclusion criteria.

Response: Thank you. We corrected this statement. It now specifies that eyes outside the range of +3.0 D to –6.0 D spherical equivalent were excluded.

Comment 2: As I understood, in your multivariable mixed effect model, you considered age and axial length. Based on important landmark studies, it is crucial that you include age, gender, IOP and CCT as well. Please include those variables and report the results of that model.

Response: Thank you for this important comment. We performed additional univariable analyses including gender, intraocular pressure, and central corneal thickness. None of these variables showed significant associations with FAZ parameters (p > 0.05). However, given our sample size, we opted not to include them in the multivariable mixed-effects model to avoid overfitting. Therefore, only age and axial length were retained in the multivariable model. This information has been clarified in the Methods and Results sections.

Comment 3: Figures 6-8 are box plots and please correct the figure legend (you incorrectly mentioned scatter plot).

Response: Thank you. We corrected the legends of Figures 6–8 to indicate box plots.

Comment 4: The discussion is comprehensive but at times overly long, with some redundancy between background context, literature comparisons, and study findings. Consider streamlining the text to focus on the most novel contributions of this study.

Response: Thank you. We revised the Discussion to remove redundancies and emphasize the novel aspects of our study.

Comment 5: Several reported correlations (e.g., FAZ area vs. mMD) are statistically significant but have low R² values. While the manuscript acknowledges this, the clinical implications of such modest effect sizes could be further clarified.

Response: : Thank you. We expanded the Discussion to address the correlations, contextualizing them within glaucoma literature and discussing their translational implications: “In our study, many of the associations demonstrated relatively low coefficients of determination (R²< 0.25). This finding is not unexpected in glaucoma research[42]. Structure-function relationships in glaucoma are inherently complex and influenced by multiple biological and methodological factors, including variability in VF testing, inter-individual anatomical differences, and age-related changes[43–48]. Low R2 values therefore do not necessarily indicate that the associations lack clinical significance. Rather, they suggest that FAZ metrics should not be interpreted in isolation but in conjunction with other structural and functional parameters[49]. Importantly, even modest correlations may provide incremental diagnostic or prognostic value when integrated into multimodal assessment, particularly for longitudinal monitoring where small but consistent changes may indicate disease progression[50,51]. Thus, while our results do not justify clinical implementation of FAZ parameters as stand-alone biomarkers, they reinforce their potential role as complementary metrics within a broader multimodal glaucoma evaluation framework.”

Comment 6: The comparisons with previous studies are detailed, but in some cases they read as a list of findings rather than a synthesis. You might consider summarizing the key differences between your study and prior work in a table.

Response: Thank you. We created a table (Table 3) summarizing key diferences between our study and prior studies.

Comment 7: The finding that age independently affects FAZ perimeter and circularity is important, but the discussion could benefit from more elaboration on the biological mechanisms behind these changes and whether they may confound glaucoma staging.

Response: We expanded the Discussion to elaborate on possible biological mechanisms and how age-related changes may confound glaucoma staging.

Conclusion

We thank both reviewers for their insightful comments. We believe these revisions have strengthened the manuscript considerably.

---

## [Decision Letter · Decision Letter 1]

15 Sep 2025

High-density perimetry in the assessment of foveal avascular zone and macular structure in glaucoma

PONE-D-25-37909R1

Dear Dr. Paranhos Jr,

We’re pleased to inform you that your manuscript has been judged scientifically suitable for publication and will be formally accepted for publication once it meets all outstanding technical requirements.

Kind regards,

Shinji Kakihara, M.D.,Ph.D.

Academic Editor

PLOS ONE

Additional Editor Comments (optional):

Thank you for the revision. Authors adequately addressed the reviewers' concerns.

Reviewers' comments:

Reviewer's Responses to Questions

**Comments to the Author**

1. If the authors have adequately addressed your comments raised in a previous round of review and you feel that this manuscript is now acceptable for publication, you may indicate that here to bypass the “Comments to the Author” section, enter your conflict of interest statement in the “Confidential to Editor” section, and submit your "Accept" recommendation.

Reviewer #2: All comments have been addressed

2. Is the manuscript technically sound, and do the data support the conclusions?

Reviewer #2: Yes

3. Has the statistical analysis been performed appropriately and rigorously? 

Reviewer #2: Yes

4. Have the authors made all data underlying the findings in their manuscript fully available?

Reviewer #2: Yes

5. Is the manuscript presented in an intelligible fashion and written in standard English?

Reviewer #2: Yes

6. Review Comments to the Author

Reviewer #2: All comments have been addressed accordingly, which helped improved your study and made more scientific and understandable. Thank you!

7. PLOS authors have the option to publish the peer review history of their article (what does this mean? ). If published, this will include your full peer review and any attached files.

**Do you want your identity to be public for this peer review?** For information about this choice, including consent withdrawal, please see our Privacy Policy .

Reviewer #2: **Yes: ** Vahid Mohammadzadeh

---

## [Editor Report · Acceptance letter]

PONE-D-25-37909R1

PLOS ONE

Dear Dr. Paranhos Jr,

I'm pleased to inform you that your manuscript has been deemed suitable for publication in PLOS ONE. Congratulations! Your manuscript is now being handed over to our production team.

Kind regards,

on behalf of

Dr. Shinji Kakihara

Academic Editor

PLOS ONE